# Sodium Alginate–Aldehyde Cellulose Nanocrystal Composite Hydrogel for Doxycycline and Other Tetracycline Removal

**DOI:** 10.3390/nano13071161

**Published:** 2023-03-24

**Authors:** Xiangyu Huang, Cheng-Shiuan Lee, Katherine Zhang, Abdulrahman G. Alhamzani, Benjamin S. Hsiao

**Affiliations:** 1Department of Chemistry, Stony Brook University, 100 Nicolls Road, Stony Brook, NY 11794, USA; 2New York State Center for Clean Water Technology, Stony Brook University, Stony Brook, NY 11794, USA; 3Research Center for Environmental Changes, Academia Sinica, Taipei 115, Taiwan; 4Department of Chemistry, Imam Mohammad Ibn Saud Islamic University, Riyadh 11623, Saudi Arabia

**Keywords:** biomaterials, nanocellulose adsorbent, dialdehyde cellulose nanocrystal, antibiotic remediation, reusability, wastewater treatment

## Abstract

A novel composite hydrogel bead composed of sodium alginate (SA) and aldehyde cellulose nanocrystal (DCNC) was developed for antibiotic remediation through a one-step cross-linking process in a calcium chloride bath. Structural and physical properties of the hydrogel bead, with varying composition ratios, were analyzed using techniques such as BET analysis, SEM imaging, tensile testing, and rheology measurement. The optimal composition ratio was found to be 40% (SA) and 60% (DCNC) by weight. The performance of the SA–DCNC hydrogel bead for antibiotic remediation was evaluated using doxycycline (DOXY) and three other tetracyclines in both single- and multidrug systems, yielding a maximum adsorption capacity of 421.5 mg g^−1^ at pH 7 and 649.9 mg g^−1^ at pH 11 for DOXY. The adsorption mechanisms were investigated through adsorption studies focusing on the effects of contact time, pH, concentration, and competitive contaminants, along with X-ray photoelectron spectroscopy analysis of samples. The adsorption of DOXY was confirmed to be the synergetic effects of chemical reaction, electrostatic interaction, hydrogen bonding, and pore diffusion/surface deposition. The SA–DCNC composite hydrogel demonstrated high reusability, with more than 80% of its adsorption efficiency remaining after five cycles of the adsorption–desorption test. The SA–DCNC composite hydrogel bead could be a promising biomaterial for future antibiotic remediation applications in both pilot and industrial scales because of its high adsorption efficiency and ease of recycling.

## 1. Introduction

Antibiotics are synthesized chemotherapeutic agents used for medical prevention or treatment of bacterial infections. Over recent decades, the global use of antibiotics has boomed exceedingly due to increased antibiotic consumption in human medicine and other commercial activities [1]. It has been reported that the consumption of antibiotics in the livestock industry alone reached 63,151 tons in 2010 and was estimated to grow by 67% by 2030 [2]. The overuse of antibiotics, along with unprocessed discharge and unassimilated excretion by humans and animals, has caused elevated accumulations of antibiotics in both manufactured environments (i.e., sewage and wastewater treatment plants (WWTPs)) and natural environments (i.e., terrestrial, freshwater, and marine) [3]. Long-term exposure to antibiotics can promote the emergence of antibiotic-resistance genes (ARGs) and antibiotic resistance-carrying bacteria (ARBs), leading to reduced effectiveness of existing antibiotics [1]. Unfortunately, due to the lack of policies and regulations, current WWTPs have not been specifically designed to manage the antibiotic and ARG pollution in wastewater [4]; therefore, it is essential to develop sustainable, efficient and cost-effective technologies to facilitate the removal of antibiotics from polluted water.

Tetracyclines are a group of broad-spectrum antibiotics widely used in animal husbandry to treat bacterial or protozoan infections and as food additives for growth promotion [5]. The simplest tetracycline is 6-demethyl-6-deoxytetracycline, which represents the backbone structure of all tetracyclines. Tetracyclines have three active pK_a_ sites, as shown in Figure 1. In this figure, site 1 (right) is the conjugated trione system (including a primary amine group), which is acidic in nature with a pK_a_ range of 2.8–3.4. Site 2 (left) is a slightly basic conjugated phenolic enone system with a pK_a_ range of 7.2–7.8. Site 3 (top) contains a strong alkaline (dimethylamino group) with a high pK_a_ range of 9.1–9.7 [6]. Due to the combined effects of the three pK_a_, tetracyclines present various ionic forms (cationic, zwitterionic, or anionic) at different pH conditions [7]. This complex amphoteric property of tetracyclines may cause difficulty in developing an efficient adsorbent, as the adsorption capacity might be significantly affected by pH. Moreover, the pK_a_ values of one tetracycline could be different from another tetracycline due to different locations of the functional groups.

The major compounds of the tetracycline family include tetracycline (TC), oxytetracycline (OXY), chlortetracycline (CTC), and doxycycline (DOXY). Their structure and pK_a_ values are summarized in Table 1 [8]. Biodegradation of tetracyclines is challenging due to their stable structures (including benzene rings, -NH_2_ and -CH_3_ groups). Thereby, the discharge of tetracyclines into water bodies poses risks to our ecosystem (i.e., adverse effects, drug interactions, and bacterial resistance), and requires appropriate controls and regulations [9]. In this study we selected doxycycline, the first and most commonly used semisynthetic antibiotic in the tetracycline family, as the primary target pollutant.

Although adsorption technology has been recognized as an easy and potent method for removing and separating organic compounds, it has not been widely explored for antibiotic removal [3]. Some recently reported adsorbents for tetracyclines include activated carbon (AC) [10], biochar (BC) [11,12], carbon nanotubes (SWCNT) [13], multiwalled carbon nanotubes (MWCNT) [14], clay mineral (bentonite) [9], graphene oxide (GO) [15], humic substances, and polymer resin [16]. The corresponding reported results suggested that the adsorption efficiencies of these materials are usually dependent on the pH, ionic strength, and the properties of adsorbents, such as the specific surface area (SSA), porosity, pore size, and surface functional groups, etc. Ahmed et al. [17] reviewed literature about different adsorbents for tetracycline removal by evaluating their adsorbent-to-solution distribution coefficient (K_d_) values. According to their report, the preferred adsorbents follow the trend SWCNT > graphite > MWCNT = AC > bentonite = humic substance = clay minerals, while the trend for the materials costs is BC < AC < ion exchange resin < MWCNT < SWCNT [17]. It can be observed that the highly effective adsorbents are usually more expensive. Therefore, future research challenges for antibiotic removal fall on developing adsorbents of high efficiency with decreased costs in raw materials, production, and regeneration processes.

Sodium alginate (SA), a natural biopolymer extracted from brown algae and seaweed, is a good candidate as an adsorbent material for removing tetracyclines in water due to its nontoxicity, biodegradability, biocompatibility, low cost, and abundant availability [18]. However, alginate materials usually need physicochemical modifications (i.e., grafting, copolymerization, and hydrogel fabrication) to improve their properties and durability and incorporate new functional groups for specific adsorption applications [19,20]. Recently, research on alginate hydrogel compositing with other sustainable biomaterials (cellulose) for water treatment has attracted significant attention. Hu et al. [19] prepared a carboxylated cellulose–alginate hydrogel bead material for Pb(II) adsorption and reported it had excellent adsorption capacity in continuous recycling tests. Tam’s research group developed anionic cellulose nanocrystal incorporated alginate hydrogel materials and demonstrated their adsorption efficiency for cationic dye in both batch and fixed-bed column adsorption studies [18,21]. However, the adsorption performance of alginate and cellulose composite hydrogel materials to remove antibiotic pollutants from water has never been reported.

Dialdehyde nanocellulose is a promising material that exhibits a wide range of chemical reactivity owing to the aldehyde functional groups on its backbone [22]. However, the production of dialdehyde cellulose via periodate oxidation of cellulose entails significant defibrillation [23], which impedes the recovery and thus practical applications of the material in wastewater treatment. To circumvent this limitation and to exploit the functionality of dialdehyde cellulose for antibiotic remediation, the present study investigated the fabrication and performance of a novel adsorbent composed of a sodium alginate hydrogel bead embedded with aldehyde cellulose nanocrystals (DCNC). The adsorbent was evaluated for its efficacy in removing tetracyclines using doxycycline (DOXY) as a model compound. The specific topics investigated in the current study were: (i) the preparation of SA–DCNC composite hydrogel with high efficiency for tetracycline adsorption, such as the effect of SA–DCNC component weight ratio on the hydrogel’s formation and properties; (ii) the examination of the effect of various adsorption factors on the adsorption capacity of SA–DCNC hydrogel beads for DOXY, such as contact time, initial DOXY concentration, temperature, pH, and competition between multiple drugs; and (iii) the investigation of adsorption behavior using kinetic and isotherm models, the characterization of structure change of the material after adsorption, as well as the interpretation of the adsorption mechanism of DOXY onto SA–DCNC hydrogel.

## 2. Experimental

### 2.1. Materials

Cellulose nanocrystal (CNC) aqueous slurry (~12 wt.%) was purchased from the University of Maine. Sodium periodate (NaIO_4_, 99%), sodium alginate (100%), tetracycline hydrochloride (C_22_H_25_ClN_2_O_8_, >95%), chlortetracycline hydrochloride (C_22_H_23_ClN_2_O_8_·HCl, >95%), oxytetracycline hydrochloride (C_22_H_25_ClN_2_O_9_, 100%), doxycycline hydrochloride (C_22_H_25_ClN_2_O_8_, >95%), and calcium chloride dihydrate (CaCl_2_·H_2_O, >95%) were all purchased from Fisher Scientific, Rochester, NY, USA. All chemicals and materials were used without further treatment.

### 2.2. Fabrication of Alginate–Cellulose Nanocrystal Composite Hydrogel Beads

#### 2.2.1. Preparation of Dialdehyde Cellulose Nanocrystal (DCNC)

In a typical synthesis, 40 mL of diluted CNC suspension (2.5 wt.%) was prepared and heated to 55 °C. Subsequently, 2.139 g of NaIO_4_ was added to initiate the reaction, where the mixture was stirred in the dark for 24 h to complete the reaction. Afterwards, 1 mL of ethylene glycol was added to the mixture to quench the remaining periodate. The resulting DCNC suspension was transferred to a dialysis bag and dialyzed in a fresh deionized (DI) water bath for about 1 week until the dialysate conductivity was below 10 μS/cm. This sample was then centrifuged to eliminate some formed gel and impurities.

#### 2.2.2. Preparation of SA–DCNC Hydrogel Beads

The SA–DCNC hydrogel beads were prepared by an ionotropic gelation method using CaCl_2_ as the cross-linking bath. First, sodium alginate (SA) was dissolved in DI water (2.5 wt.%) with mild stirring for over 2 h to obtain a homogeneous solution. Then, an appropriate amount of DCNC suspension was added to the SA solution under stirring to obtain the homogeneous SA–DCNC mixture. Four mixtures with various SA–DCNC ratios (100/0, 80/20, 60/40, and 20/80) were obtained to investigate the composition effects on beads. The details of each sample composition are summarized in Table 2.

The prepared SA–DCNC mixture was loaded into a syringe and injected through a tubing (ID 0.75 mm) using a syringe pump at a flow rate of 0.5 mL min^−1^ into 100 mL of CaCl_2_ solution. The tubing was fixed at 10 cm height from the bath-liquid surface while the CaCl_2_ bath was under gentle stirring. The beads were allowed to cross-link in the CaCl_2_ bath for 20 min and then washed with water to remove the unbound Ca^2+^. Prepared beads were soaked in DI water overnight before being taken for characterization or adsorption testing.

### 2.3. Characterization

#### 2.3.1. Titration to Determine Aldehyde Content in DCNC

The hydroxylamine hydrochloride titration method was applied to titrate the aldehyde content on the DCNC sample [24]. Details about the calculation method of aldehyde content are described in Appendix A (Appendix A).

#### 2.3.2. Fourier-Transform Infrared (FT-IR) Spectroscopy

The chemical structures of unmodified CNC and oxidized DCNC were characterized using Fourier-transform infrared (FT-IR) spectroscopy (Nicolet iS10, Thermo Scientific, Rochester, NY, USA). Meanwhile, freeze-dried pure SA and SA–DCNC hydrogel beads were also analyzed by FT-IR to help understand the cross-linking behavior. Each sample was tested in the attenuated total reflectance (ATR) mode within the wavenumber range of 550–4000 cm^−1^, with a resolution of 0.5 cm^−1^ and an accumulation of 16 scans.

#### 2.3.3. X-ray Diffraction (XRD)

X-ray diffraction measurements were carried out on freeze-dried CNC and DCNC samples using a Rigaku Benchtop MiniFlex 600 instrument (Rigaku, Wilmington, MA, USA). The Cu-Kα radiation source (λ = 1.5406 Å) was operated at 40 kV and 15 mA. The 2θ diffraction diagrams were obtained between 5° and 60° at a scanning rate of 5° min^−1^.

#### 2.3.4. Transmission Electron Microscopy (TEM)

The morphology of CNC and DCNC were studied by transmission electron microscopy (TEM; JEOL JEM-1400, JEOL, Peabody, MA, USA) operated at 80 kV. ImageJ software (v1.51, U. S. National Institutes of Health, Bethesda, MD, USA) was then applied to measure the nonmodified/oxidized CNC width from the obtained TEM images. A minimum of 100 measurements was performed for each sample.

#### 2.3.5. Dynamic Light Scattering (DLS)

Particle size and zeta potential of CNC and DCNC were measured using a Zetasizer Nano ZS (Malvern, UK) under the following conditions: materials refractive index 1.47, material absorption 0.100, dispersant water, dispersant refractive index 1.33, viscosity 0.8872 cP, and temperature 25 ± 0.1 °C. Samples for DLS analysis were prepared by diluting the CNC or DCNC suspensions to 0.1 wt.% and sonicated overnight. The sonicated samples were subsequently filtered through the 0.45 µm PVDF microfilter before transferal to the DLS cuvette (the first 3 drops were discarded). The DLS measurements were carried out on the filtered samples within several hours after sonication. Each sample was measured in 3 cycles, and the results were averaged.

#### 2.3.6. Thermogravimetric Analysis (TGA)

TGA measurements were conducted using a Q50 thermogravimetric analyzer (TA Instruments, New Castle, DE, USA) to verify the successful incorporation of DCNC into the gel network of the SA system. About 10 mg of freeze-dried DCNC, SA bead, and SA–DCNC bead were heated from 25 °C to 800 °C at a heating rate of 10 °C min^−1^ under a nitrogen atmosphere.

#### 2.3.7. Property of SA–DCNC Composite Hydrogel

Digital photographs of SA–DCNC hydrogel beads in various compositions were recorded, and the mean diameters of the beads were determined using the ImageJ software. Meanwhile, the surface area and porosity of the SA–DCNC freeze-dried beads were analyzed by N_2_ adsorption via a BET surface area analyzer (NOVAtouch LX2, Quantachrome Instruments, Boynton Beach, FL, USA). All samples were first degassed at 90 °C for 12 h. The Brunauer–Emmett–Teller (BET) and Barrett–Joyner–Halenda (BJH) models were applied to calculate the specific surface area, pore volume, and pore radius. The surface morphology and porous structure of both pure SA and SA–DCNC beads were also examined by scanning electron microscopy (SEM; Zeiss, LEO 1550 SFEG, Dublin, CA, USA).

To understand the influence of the SA–DCNC composite ratios on the mechanical properties of the hydrogel, various SA–DCNC composite hydrogels were prepared and cut into strips (25 mm × 5 mm) for tensile strength tests using a modified Instron 4410 tensile stretching instrument. Each group of samples was measured 5 times under room temperature at a 10 mm min^−1^ stretching rate. The stress was calculated as the maximum load divided by the original cross-section area, while the strain was the difference of the original and deformed lengths of the sample divided by its original length.

SA–DCNC samples (2.5 wt.%) with various compositions were taken for rheology measurements using a modular compact rheometer (Physical MCR 301, Anton Paar, Houston, TX, USA). The steady shear viscosity of samples was recorded as a function of shear rate (1–100 s^−1^), and their dynamic rheological behavior was also studied at a frequency ranging from 1 to 100 Hz. All measurements were carried out at 25 °C.

Lastly, the surfaces of the materials before and after adsorption of doxycycline were investigated and compared using a custom PHI X-ray photoelectron spectrometer (XPS) with Al Kα1,2 (1486.6 eV) as an X-ray source.

### 2.4. Antibiotic Adsorption Studies

All adsorption experiments were conducted in triplicate. In each sample, 5 mL of antibiotic solution with 0.1 g of hydrogel beads (water content ~96 wt.%) was shaken on a shaking bed (180 rpm) in darkness at 25 ± 1 °C. In the single-drug adsorption test, the initial and final concentration of antibiotics was determined using a UV-vis spectrometer (Genesys 10S, Thermo Scientific, Rochester, NY, USA) at the maximum absorbance values of each tetracycline (276–278 nm). The premeasured standard calibration curves were used to convert the absorbance values to the corresponding concentration. Then, the amount of antibiotic adsorbed on the SA–DCNC hydrogel beads *Q_e_* (mg g^−1^) was calculated using Equation (1):(1)$$Q_{e}=\frac{\left(C_{0}-C_{e}\right)V}{m}$$
where *C*_0_ and *C_e_* (mg L^−1^) are the initial and final concentrations of antibiotics in the aqueous solution, respectively, *V* (L) is the volume of solution, and *m* (g) is the mass of the hydrogel beads used (dry mass of 0.1 g hydrogel beads was calculated based on the corresponding water content of them).

#### 2.4.1. Effects of Hydrogel Composition on Doxycycline Adsorption

To determine the optimum SA–DCNC ratio to fabricate the hydrogel beads for DOXY adsorption, the adsorption capacities of 4 hydrogel beads for DOXY were investigated and compared. All adsorption experiments were carried out with 500 mg L^−1^ DOXY solution (pH = 7) and lasted for 24 h.

#### 2.4.2. Adsorption Kinetics

DOXY adsorption kinetics were studied by adding 0.1 g of SA–DCNC-40 hydrogel beads into 5 mL of 500 mg L^−1^ DOXY solution (pH = 7). Then, DOXY concentration in the solution was determined at scheduled time intervals (0.5–24 h). DOXY uptake (*Q_t_*) was then plotted versus time (*t*) and analyzed using various kinetic models.

#### 2.4.3. Adsorption Isotherm

The adsorption isotherm of this hydrogel beads system for DOXY was investigated by adding 0.1 g of SA–DCNC-40 beads to a series of 5 mL DOXY solutions (pH = 7) with various initial concentrations (10–2000 mg L^−1^). After 24 h of adsorption under 20 °C, 35 °C, and 45 °C, respectively, the hydrogel beads were taken out, and the residual DOXY concentration of each sample was measured by UV-vis.

#### 2.4.4. Effect of pH

The effect of the initial pH on the adsorption was analyzed by adding 0.1 g of SA–DCNC-40 beads into 5 mL of 500 mg L^−1^ DOXY solution at various pH (3, 5, 7, 9, and 11). The pH values of the abovementioned DOX solutions were adjusted by using 1.0 M HCl and 1.0 M NaOH. After 24 h, the hydrogel beads were taken out, and the residual DOXY concentration of each sample was measured by UV-vis.

#### 2.4.5. Single-Drug Adsorption

To verify hydrogel beads’ adsorption behavior towards different tetracycline drugs, 0.1 g SA–DCNC-40 hydrogel beads were added to 5 mL of 500 mg L^−1^ of TC, OXY, CTC, and DOXY solutions. All solutions were adjusted to pH 7 before adsorption. After 24 h, residual tetracyclines in solutions were measured, and the adsorption capacity for each drug was calculated.

#### 2.4.6. Multidrug (Four Tetracyclines) Competitive Adsorption

To challenge the prepared hydrogel beads in a multidrug system, 0.4 g of SA–DCNC-40 hydrogel beads was added to a 20 mL solution containing an equal mixture of TC, OXY, CTC, and DOXY (all 500 mg L^−1^). After 24 h, the remaining concentration of each of the four drugs was determined by chromatographic analysis using a high-performance liquid chromatography (HPLC) system (Shimadzu Prominence Modular, Tokyo, Japan) equipped with a CBM-20A controller, two LC-20AD pumps, and a SIL-10A autosampler.

HPLC separation of the four tetracyclines was achieved on a Zorbax RP-C18 column (250 mm × 4.6 mm, 5 μm, Agilent, Santa Clara, CA, USA) at 25 °C with gradient elution at a flow rate of 1 mL/min, and the injection volume was 10 µL. The gradient began with a mobile phase consisting of 0.025 M KH_2_PO_4_ solution (A) and acetonitrile (B). The volume ratio of A:B was set at 95:5 at the beginning, then linearly changed to 60:40 in 20 min, and maintained at this condition for another 5 min. A Shimadzu SPD-M20A UV-vis photodiode array detector was operated at a fixed wavelength of 350 nm to detect the UV absorbance of TC, OXY, CTC, and DOXY.

#### 2.4.7. Reusability Test

The reusability of SA–DCNC-40 hydrogel beads after DOXY adsorption was investigated through five consecutive adsorption–desorption cycles. Specifically, 0.5 g of SA–DCNC-40 beads was added to 25 mL of 500 mg L^−1^ DOXY solution (pH = 7). After 24 h, the DOXY concentration was analyzed by UV-vis spectroscopy. Then, the recycled SA–DCNC-40 beads were soaked in fresh DI water with vigorous stirring overnight to desorb the DOXY. After washing, the regenerated beads were used for the next adsorption cycle.

## 3. Results and Discussion

The design of our SA–DCNC composite hydrogel material is illustrated in Figure 2. Initially, we prepared the dialdehyde cellulose nanocrystal (DCNC) via periodate oxidation (Figure 3a), incorporating aldehyde groups onto the material. This modification aimed to enhance the adsorption of tetracyclines through the interaction between the aldehyde groups on DCNC and the dimethylamino groups on antibiotics. Meanwhile, the electrostatic repulsive force between DCNC and SA decreased due to the lower charge density on the DCNC surface. Then, the DCNC suspension was mixed with the SA solution to form the SA–DCNC composite hydrogel beads in the CaCl_2_ bath. The effect of the SA–DCNC composition ratio on the properties of mixture, beads, and their efficiency in doxycycline adsorption was investigated. In the following sections, we discuss the characterizations of DCNC, SA–DCNC composite materials, and their adsorption properties toward the representative tetracyclines.

### 3.1. Characterizations of DCNC

Based on the titration result, the aldehyde content in DCNC was 11.52 ± 0.35 mmol g^−1^. The chemical structure of CNC and DCNC was examined and compared by FT-IR spectra, shown in Figure 3b. One can see that both CNC and DCNC show the same cellulose characteristic FT-IR peaks, i.e., O-H, C-H, and C-O stretching at around 3400 cm^−1^, 2900 cm^−1^, and 1030 cm^−1^, respectively. The peak at 814 cm^−1^ can be assigned to the S-O stretching from the sulfate groups on CNC. However, the S=O stretching at around 1350 and 1165 cm^−1^ is not noticeable, probably due to the overlaps of several other peaks in these two regions [25]. The maximum absorbance of the OH stretching vibration on CNC is seen to shift from 3334 to 3367 cm^−1^ after oxidation, which represents the intermolecular hydrogen bonding O(6)H···O(3) and intramolecular O(3)H···O(5), respectively. This shift may stand for the transformation of structure from cellulose I to cellulose II caused by periodate oxidation [26].

The crystalline structures of both CNC and DCNC samples were examined by XRD, and the results are shown in Figure 3c. The typical peaks of the cellulose I at 14.5°, 16.4°, and 22.5° associated with the (1$\bar{1}$0), (110), and (200) planes are shown in the CNC sample. In DCNC, a noticeable reduction in the diffraction intensity of the three diffraction peaks was observed, confirming the loss of crystallinity after periodate oxidation. This is consistent with the above FTIR results and the previous report [27].

The structure and morphology of CNC and DCNC samples were characterized by TEM and DLS (shown in Figure 4). One can observe that while the TEM images of CNC and DCNC show no noticeable difference, the sample width of DCNC (9.6 ± 2.6 nm) measured by TEM is slightly larger than CNC (9.1 ± 1.9 nm). The average particle size measured by DLS also follows the same trend, in which DCNC’s Z-average value was greater than CNC’s (Figure 4e,f). The unexpected size increase of DCNC can be explained by the sample’s aggregation caused by reducing the charge density. According to the zeta-potential analysis of 0.1 wt.% suspensions, the CNC sample possesses a highly negative charge of −147.3 ± 3.3 mV, while the DCNC only presents a weak surface charge of −28.0 ± 5.4 mV. During the periodate oxidation, the acidic condition may contribute to the desulfation of CNC, thus resulting in an incipient colloidal instability when the zeta-potential value is less than −30 mV [28]. The weak negative surface charge of DCNC is beneficial for the following SA–DCNC composite material fabrication, as the repulsive force between SA and DCNC is reduced, which may facilitate formation of a more strongly connected three-dimensional network.

### 3.2. Formulation and Characterization of SA–DCNC Hydrogel

Hydrogel materials with an interconnected porous structure and high specific surface area are highly desirable to facilitate water absorption while maintaining mechanical strength [29]. In order to explore the optimum formulation, hydrogel beads were prepared in four weight ratios of SA–DCNC to investigate their individual properties and contribution to the adsorption of the representative drug (DOXY).

The bead diameter (D_B_), BET specific surface area (SSA), BJH pore volume (V_P_), and BJH mean pore radius (r_P_) for various SA–DCNC hydrogel bead samples were measured and are summarized in Table 2. In general, increasing the composition of DCNC in the hydrogel produces hydrogel beads with a smaller diameter and higher water content; however, the shape of the bead becomes unstable when the SA–DCNC ratio reaches 20/80. Compared with the neat alginate hydrogel bead (SA), the SSA of SA–DCNC composite samples improved by over 50%, and notably the SSA of SA–DCNC-40 increased the most (117.7%). It was expected that the DCNC introduced into the system would help the hydrogel to form a more porous structure, as confirmed by the increase of the samples’ pore volume, thereby contributing to an increase in the total surface area. It is worth mentioning that the mean pore size decreased with the DCNC component increase, which is probably attributable to partial pore blockage caused by the DCNC [30]. Thus, though incorporating DCNC into hydrogel can increase the number of active sites and surface area for adsorption, too much DCNC could block the tiny internal pores and hinder mass transfer.

To further study the DCNC component’s effect on the porous structure of hydrogel beads, the surface and cross-section morphologies of freeze-dried SA and SA–DCNC-40 samples were observed by SEM (shown in Figure 5). As shown, the surface of the neat SA hydrogel is very smooth, while the SA–DCNC-40 surface shows many irregular folds. The roughness of the composite hydrogel surface can effectively increase the surface area available for adsorbate attachment [21]. Moreover, the neat SA bead exhibits a porous morphology with a random stratified structure (Figure 5b), which is similar to a previous report [29]. However, after incorporating DCNC into the hydrogel, it is interesting to observe that the SA–DCNC-40 hydrogel developed a three-dimensional network with interconnected pores. The SEM images also indicate that no apparent self-aggregation of cellulose nanocrystals was formed, and the SA–DCNC was combined homogeneously in the system.

Figure 6a shows the performance of various SA–DCNC hydrogel samples in doxycycline adsorption. As the DCNC weight ratio increased from 0 to 80%, the DOXY adsorption efficiency increased 2.5 times. This indicates that the increase in DCNC component in hydrogel can efficiently increase the adsorption capacity for doxycycline, due to the contribution of the active sites from the modified nanocellulose material with abundant aldehyde groups. All hydrogel samples used in this study were in the never-dried form. It was found that the dried hydrogel beads exhibited much lower adsorption capacity compared to the wet ones. This could be attributed to the irreversible loss of porosity during the drying process, thus increasing the barrier to the diffusion of the drug molecules to the internal adsorption binding sites [31].

The mechanical property is also an essential parameter for a composite hydrogel material applied as an adsorbent in the water purification process. Figure 6b shows the tensile stress–strain curves of the hydrogel strip samples with different SA–DCNC weight ratios in a wet state. The highest mechanical performance was obtained by SA–DCNC-20, of which the maximum tensile strength (243.3 kPa) increased by a factor of 1.41 compared to the neat SA. It can be inferred that the rigid cellulose nanocrystals in the hydrogel network had effectively helped to endure and transfer the stress of the system, thus exhibiting a significant nano-reinforcement effect. However, if superfluous DCNC were incorporated in the hydrogel, the reinforcing effect would decrease due to the weak cross-linking formed and the self-aggregation of nanocrystals (i.e., SA–DCNC-80). This result is consistent with the previously reported result based on a TEMPO-mediated oxidized CNC [29].

This study also investigated the rheological behavior of the various SA–DCNC mixture suspensions (2.5 wt.%) to decide the optimum hydrogel formulation. First, from the curves of the viscosity versus steady shear rate shown in Figure 6c, the neat SA sample exhibited the highest viscosity, and the sample’s viscosity decreased significantly with the increase in DCNC weight ratio from 0 to 80%. Meanwhile, the pure DCNC suspension (2.5 wt.%) behaved like a Newtonian fluid, and zero viscosity was observed (not shown in the graph). Therefore, to ensure the SA–DCNC mixture’s favorable and stable gelation, superfluous low-viscosity DCNC components should be avoided. The pure SA sample displayed a Newtonian plateau at a low shear rate region, which is typical for polymer solutions [32]. As the shear rate increased, all samples exhibited a similar shear-thinning behavior induced by the alignment of both alginate chains and DCNC rods under high shear rates. This observation suggests the presence of homogeneous DCNC–SA dispersion, where no apparent aggregation or entanglement was formed before the addition of CaCl_2_.

Frequency sweep tests were done, and the storage modulus (G′) and loss modulus (G″) as a function of angular frequency for various samples are shown in Figure 6d. It was seen that the SA–DCNC mixtures exhibited liquid-like behavior (G′ < G″) for all cases, indicating that the hydrogen bonds formed between the carboxylate groups on alginate and the hydroxyl groups on DCNC were too weak to maintain stable physical gelation under this concentration [33]. Pinto et al. reported that the solid-like character would only become predominant when the CNC concentration is above 4 wt.% [32].

Overall, after comprehensively considering the effects of various SA–DCNC weight ratios on all of the above results, the hydrogel beads with a 60/40 (*w*/*w*) SA to DCNC ratio (SA–DCNC-40) were chosen as the optimum formulation and used for further characterization and adsorption studies.

To further investigate the intermolecular interaction between SA and DCNC composites and the ionotropic interaction during the cross-linking process, the chemical structures of DCNC, SA, and SA–DCNC freeze-dried beads were investigated by FT-IR and are compared in Figure 7a. The characteristic peaks of SA located at 1594 cm^−1^ and 1408 cm^−1^ correspond to the symmetric and asymmetric stretching vibration of carboxylate groups (-COO^−^), respectively [34]. In the spectrum of SA–DCNC, it can be observed that two -COO^−^ shifted to 1597 cm^−1^ and 1418 cm^−1^, respectively. The shifts toward higher wavelengths could be induced by the ionic cross-linking between COO^−^ and Ca^2+^. The intensity of the O-H band at around 3334 cm^−1^ shows an evident increase in the SA–DCNC sample compared with the pure DCNC or SA spectrum, which confirms the formation of hydrogen bonds between SA and SA DCNC in the SA–DCNC composite hydrogel. The adsorption bands of aldehyde groups at 1730 cm^−1^ from DCNC could not be observed in SA–DCNC. Overall, the FT-IR test suggests good molecular affinity between SA and DCNC and the potential occurrence of internal chemical interactions [35].

The thermal behavior of the DCNC, SA, and SA–DCNC composite samples was assessed by TGA, and the thermograms are shown in Figure 7b,c. All three samples exhibited a first weight-loss step at 40–100 °C, indicating complete moisture removal. Next, the DCNC sample displayed a primary weight-loss step at 125–350 °C with a maximum decomposition rate at 231.2 °C. Cellulose material has been reported to exhibit a decomposition temperature between 260 °C and 360 °C [36]. The poor thermal stability and broad degrading temperature range of DCNC compared with the native cellulose is mainly attributed to the increase in amorphous content after periodate oxidation [37]. In the SA sample, a sharp peak at 191.3 °C followed by a broad peak at 235.8 °C (200–300 °C) corresponds to the principal decomposition of the alginate polymer chain [38]. The final complete degradation step of SA occurred at 685.9 °C (600–750 °C), in which the CaCO_3_ and other carbonized materials were formed, resulting in 15.7% residual mass [39]. The composite SA–DCNC cross-linked sample exhibited similar decomposition stages with SA. However, the SA–DCNC sample showed increased thermal stability represented by a higher residual mass (25%), induced by incorporating Ca^2+^ ions and the formation of the stable cross-linked “egg-box” complex [40].

### 3.3. Adsorption of Tetracycline Antibiotics on SA–DCNC-40 Hydrogel Bead

#### 3.3.1. Kinetic Study

The adsorption kinetics of representative tetracycline drug DOXY on SA–DCNC-40 hydrogel beads were evaluated. As shown in Figure 8a, the adsorption amounts of DOXY improved gradually and achieved 90% of the equilibrium amount after 8 h. The adsorption kinetic data were fitted with pseudo-first-order, pseudo-second-order, and intraparticle diffusion kinetic models. All fitting parameters are listed in Table 3. The pseudo-second-order model is found to fit the experimental data best with the highest correlation coefficient R^2^ (0.9999), which indicates that the process of DOXY adsorption is chemisorption-controlled [41]. In order to determine the involvement of pore diffusion in the adsorption mechanism, the intraparticle diffusion model was also investigated. The intraparticle diffusion plot shows that three adsorption steps with different slopes were achieved with high R^2^ (0.9610–0.9999), confirming that both boundary layer diffusion and internal pore diffusion effects are involved in the DOXY adsorption by the hydrogel beads [42]. The three steps correspond to the diffusion process of DOXY molecules into the macropores of hydrogel beads, then into the mesopores (2–50 nm), and lastly into the micropores (<2 nm), respectively [41]. The value of *K_id_* in the three steps decreased in sequence, which indicates that the diffusion rate decreased with the increase of the boundary layer thickness in the system.

#### 3.3.2. Isotherm Study

Batch isotherm experiments were carried out to investigate the effects of initial concentration and temperature on the adsorption efficiency against DOXY by SA–DCNC-40 (Figure 9a). The adsorption isotherm data were fitted with both Langmuir and Freundlich models, and the plots are shown in Figure 9b,c. The results show that the adsorption data fit well with the Freundlich model (summarized in Table 4), while the Langmuir isotherm plots present poor fitting. This phenomenon confirms that multilayer chemical adsorption controls the DOXY adsorption by SA–DCNC hydrogel and the active sites on the adsorbent are heterogeneous. Meanwhile, the values of 1/n under all three temperatures are below 1, implying the DOXY adsorption process was favorable in all cases [43]. One can observe that the maximum adsorption capacity for DOXY increased with the temperature increase in the system, which were 421.5, 556.6, and 594.6 mg g^−1^ at 20 °, 35 °, and 45 °C, respectively. Many adsorption studies related to tetracyclines have reported similar observations, i.e., that adsorption is favorable at higher temperatures [44,45].

To present the excellent performance of our SA–DCNC hydrogel sample for doxycycline removal, we compared the *Q_m_* values at 25 °C in this work with other recently reported adsorbents (Table 5). We can conclude that the SA–DCNC hydrogel synthesized in this study exhibits superior adsorption capacity toward DOXY over most biomass-based materials. It also displays competitiveness against other mineral or polymeric adsorbents in terms of cost-effectiveness and environmental friendliness.

#### 3.3.3. Effect of pH

As pH can affect the molecular form of DOXY in the aqueous solution and significantly alter the surface functionalities and physical properties of the hydrogel materials, it is vital to investigate the pH effect on the performance of SA–DCNC samples for DOXY adsorption. In Figure 10a, it is interesting to note that SA–DCNC-40 exhibits brilliant adsorption efficiencies for DOXY under alkaline conditions (pH 9–11), while its capacity dramatically decreases when in acidic conditions (pH 3–5). This result suggests that the adsorption of DOXY by hydrogel in this study is highly pH-dependent. The zeta potential of SA–DCNC mixed suspension was investigated under different pH (Figure 10b). The mixture remained negatively charged under the pH range tested (3–11), and its zeta potential decreased with increasing pH, representing a higher negative charge density under basic conditions due to the deprotonation of -COO^−^ and -OSO_3_^−^. Moreover, as DOXY has three acid dissociation constants (pK_a1_ = 3.4, pK_a2_ = 7.7, pK_a3_ = 9.3), it exists in the cationic (DOXY^+^), zwitterionic (DOXY^±^), and anionic (DOXY^−^) forms under acidic, moderately acidic to neutral, and alkaline conditions, respectively [49]. At pH 3–5, the surface charge density of the hydrogel beads is relatively low, while a significant number of H^+^ ions will compete with the DOXY^+^ for the active sites on hydrogel to form a tight network via hydrogen bonding, and thus lead to low adsorption capacities [9]. In the neutral pH environment, though the DOXY molecule overall presents an electrically neutral state, the dimethylamino groups on it exhibit a positive charge, which enhance the electrostatic interaction between DOXY and the negatively charged SA–DCNC hydrogel and result in higher adsorption performance in this pH range. At the high pH range of 9–11, although the electrostatic repulsion force existed between the adsorbent and adsorbate due to both of their negatively charged surfaces, the adsorption capacity still presented a significant increase. This interesting phenomenon could be explained as follows. The anionic SA–DCNC hydrogel is pH-sensitive, which would swell dramatically at high pH due to the complete ionization of the acidic groups (-COO^−^ and -OSO_3_^−^) on it (shown in Figure 10c), resulting in a loose hydrogel network and higher mass penetration [50].

#### 3.3.4. Competitive Adsorption Test

To emphasize the excellent performance of the SA–DCNC hydrogel sample for doxycycline adsorption, the adsorption tests for four tetracyclines (TC, DOXY, OXY, CTC) under single- and multidrug environments were further investigated and analyzed by HPLC. When only a single drug was presented, the hydrogel adsorption capacities for OXY, TC, CTC, and DOXY were 189.1, 165.5, 309.2, and 402.5 mg g^−1^, respectively (Figure 11b). When a mixed solution containing four drugs was tested by SA–DCNC-40 beads, it was found that the adsorption capacities for OXY, TC, and CTC decreased significantly (by 48.1%, 46.9%, and 48.3%, respectively), while the *Q_e_* for DOXY only dropped 17.3% at the equilibrium state. Both single- and multidrug tests confirmed that the SA–DCNC hydrogel exhibits outstanding selective adsorption capacity toward DOXY compared with the other tetracyclines in the study. The different stereochemical configurations of tetracyclines may explain the preferential removal toward DOXY. The functional groups and benzene rings in doxycycline molecules are located within the same plane or point toward the same side, while the functional groups in other tetracyclines extend to both the inner and outer sides of the benzene ring plane [12]. Thus, the adsorption and diffusion behavior of DOXY was favored due to lower steric hindrance.

#### 3.3.5. Reusability Test

The reusability of adsorbents is a significant factor when considering practical water treatment applications. The reusability of SA–DCNC-40 after DOXY adsorption was evaluated and is summarized in Figure 11c. To prevent the beads’ structure from getting destroyed by organic solvent, DI water was adopted for mild desorption with stirring overnight. After five cycles of the adsorption–desorption test, the removal efficiency for DOXY remained over 80% of the initial efficiency, indicating the SA–DCNC hydrogel sample’s good recycling capacity. This result also suggests that loading the functionalized cellulose materials onto the alginate hydrogel platform effectively solves the difficulties of recycling nanosorbents after adsorption. Moreover, the efficient and low-cost desorption method makes it more feasible to regenerate and reuse the hydrogel beads in water treatment applications.

### 3.4. Adsorption Mechanism

To further confirm the possible adsorption mechanism for DOXY by SA–DCNC hydrogel beads, the SA–DCNC-40 sample before and after adsorption of DOXY was further characterized by XPS. As shown in the broad-scan XPS spectra (Figure 12a), except for identified C, O, and Ca elements in SA–DCNC, a new peak associated with the N element could be identified in the SA–DCNC–DOXY. Before adsorption, the high-resolution C 1s spectrum of SA–DCNC was deconvoluted into four peaks at 284.8, 286.4, 288.0, and 289.5 eV (Figure 12b), corresponding to C-C/C-H, C-O/C-O-C, C=O/O-C-O, and O-C=O bonds, respectively [51]. After adsorption, the C 1s spectrum of SA–DCNC was deconvoluted into five peaks at 285, 285.9, 286.3, 287.8 and 289.6 eV, where the new peak at 285.9 eV can be assigned to the C-N groups in DOXY [52] and the shifts of the existing peaks suggest interaction between DOXY and SA–DCNC. The high-resolution O 1s spectrum of SA–DCNC revealed three peaks at 531.4, 532.7, and 533.6 eV (Figure 12c), which can be assigned to the binding energy of C=O, -O-H, and C-O-C, respectively [53]. After DOXY adsorbed onto SA–DCNC, the area of the C=O peak increased from 19.3% to 30.3%, which can be attributed to the carbonyl groups presented in the loading DOXY molecules. The high-resolution N 1s spectrum of SA–DCNC- (Figure 12d) was deconvoluted into three peaks at 396.3, 398.9, and 400.5 eV, corresponding to N=C, -CONH_2_, and N-C, respectively [46,54]. The -CONH_2_ and N-C peaks were from the adsorbed DOXY molecule, while the N=C peak confirms the formation of imine groups between DCNC and DOXY. The Ca 2p XPS spectrum of SA–DCNC before adsorption was deconvoluted into two peaks at 347.5 eV (Ca 2p_3/2_) and 351.0 eV (Ca 2p_1/2_) [55]. After DOXY adsorption, the two peaks shifted to 347.2 eV and 350.7 eV, respectively, and their intensities showed a significant decrease. This change may be induced by the interaction between Ca^2+^ and the oxygen or nitrogen atoms on doxycycline molecules, followed by an electron transfer from oxygen or nitrogen to Ca^2+^ or the ion exchange during the adsorption process [19,53].

From the adsorption and characterization results, one can realize that the adsorption mechanism for DOXY on SA–DCNC hydrogel beads is complicated. A schematic illustration has been drawn to propose all the possible adsorption mechanisms (Figure 13). In brief, several possible adsorption mechanisms can be summarized: (i) the hydroxyl and carboxylate groups on SA and DCNC can act as the linking groups for the amino and hydroxyl groups on DOXY to form strong hydrogen bonds; (ii) a relatively weaker interaction (than hydrogen bonding) can be formed between the aldehyde groups on DCNC and the dimethylamino groups on DOXY [56]; (iii) within a specific pH range, the dimethylamino group on DOXY can be protonated, and hence electrostatic attraction between the DOXY and anionic SA–DCNC hydrogel can occur [57]; and (iv) the mesopores in the hydrogel can play a role in accommodating the DOXY molecules, while the pore-filling diffusion phenomenon would also favor the adsorption [9].

## 4. Conclusions

In this study, by cross-linking an alginate–aldehyde cellulose nanocrystal blending suspension in a calcium chloride bath, we fabricated SA–DCNC hydrogel beads with porous structure and controlled properties for efficient antibiotic remediation via quick adsorption. The structural and physical properties of the composite SA–DCNC hydrogel beads with different compositions were explored thoroughly via various characterization techniques, i.e., BET, SEM, tensile test, rheology measurement, etc. Experimental results indicated that SA–DCNC composite hydrogel samples show a higher adsorption capacity toward DOXY (1.6–2.5 times) than the neat SA hydrogel. With a chosen weight ratio (40:60) of SA:DCNC, the SA–DCNC-40 hydrogel sample exhibited excellent adsorption performance towards DOXY in single- and multidrug systems under neutral to high pH conditions. The maximum adsorption capacity for DOXY at pH 7 was 421.5 mg g^−1^ at 25 °C, and 594.6 mg g^−1^ at 45 °C. The isotherm data fitted well with the Freundlich model, while the kinetic data fitted both pseudo-second-order and intraparticle diffusion models. The adsorption studies and XPS analysis of samples before and after adsorption have elucidated that the adsorption mechanism of DOXY on SA–DCNC hydrogel beads is mainly caused by the synergetic effects of chemical reaction, electrostatic interaction, hydrogen bonding, and pore diffusion/surface deposition. Due to the platform of SA hydrogel, our SA–DCNC adsorbent can be easily regenerated and reused. Based on the simple and cost-efficient fabrication approach, sustainable materials with high efficiency, and ease in use and regeneration, the SA–DCNC hydrogel beads in this study could be envisioned as having potential value for the removal of DOXY and other similar antibiotics for future water purification applications on both pilot and industrial scales.

## Figures and Tables

**Figure 1 nanomaterials-13-01161-f001:**
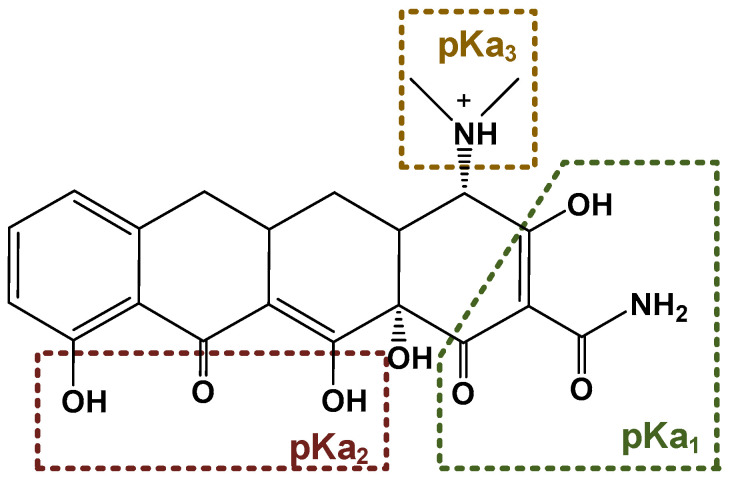
The backbone structure of the tetracycline family of antibiotics and its three pK_a_ sites.

**Figure 2 nanomaterials-13-01161-f002:**
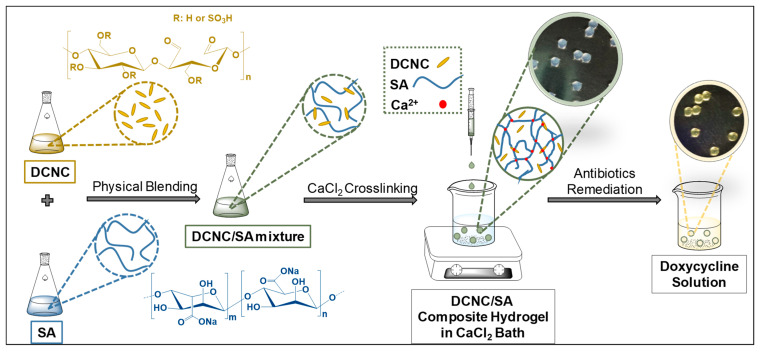
Schematic illustration of the fabrication of SA–DCNC composite hydrogel beads and doxycycline remediation.

**Figure 3 nanomaterials-13-01161-f003:**
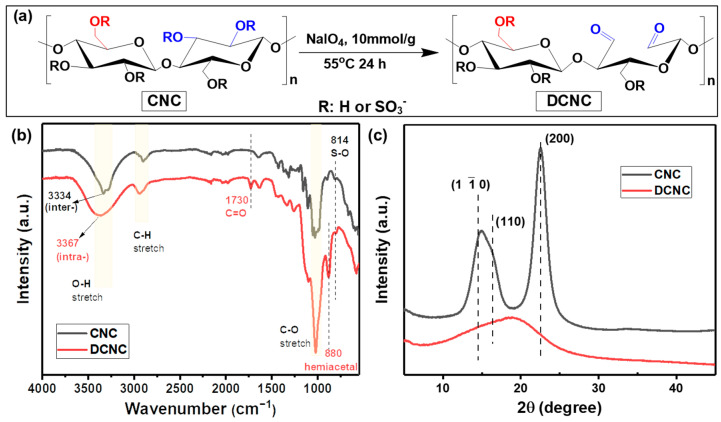
(**a**) Synthesis of dialdehyde cellulose nanocrystal (DCNC). (**b**) FT-IR spectra and (**c**) XRD patterns of CNC and DCNC.

**Figure 4 nanomaterials-13-01161-f004:**
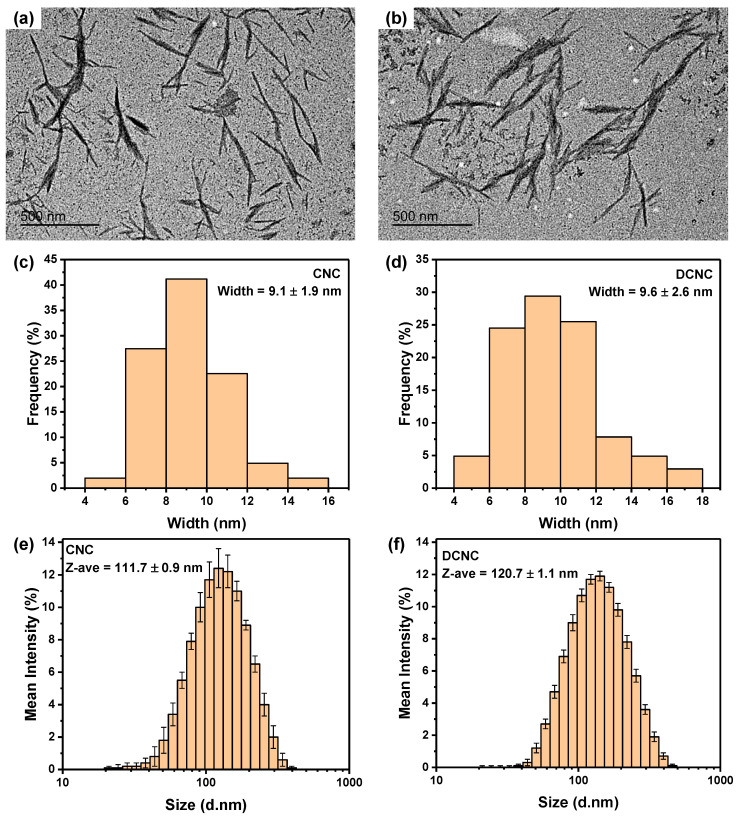
TEM images and the corresponding histograms of width distribution from TEM images: (**a**,**c**) CNC and (**b**,**d**) DCNC; size distribution histogram of (**e**) CNC and (**f**) DCNC measured by DLS.

**Figure 5 nanomaterials-13-01161-f005:**
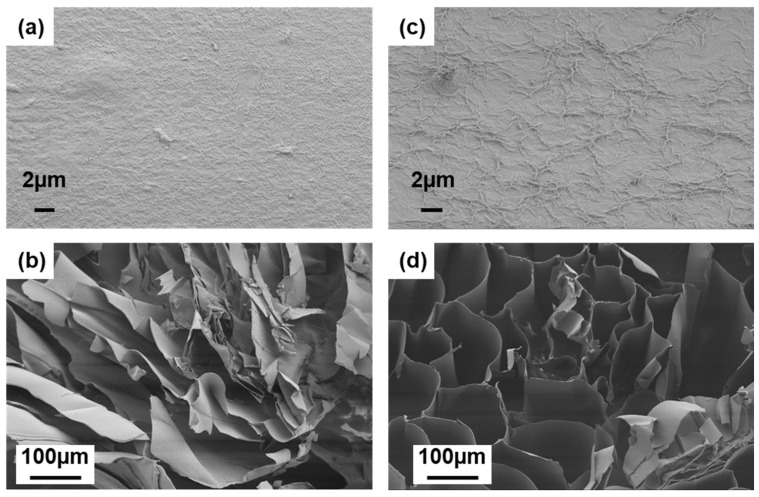
The surface and cross-sectional SEM images of (**a**,**b**) SA and (**c**,**d**) SA–DCNC-40 hydrogel bead at different magnifications.

**Figure 6 nanomaterials-13-01161-f006:**
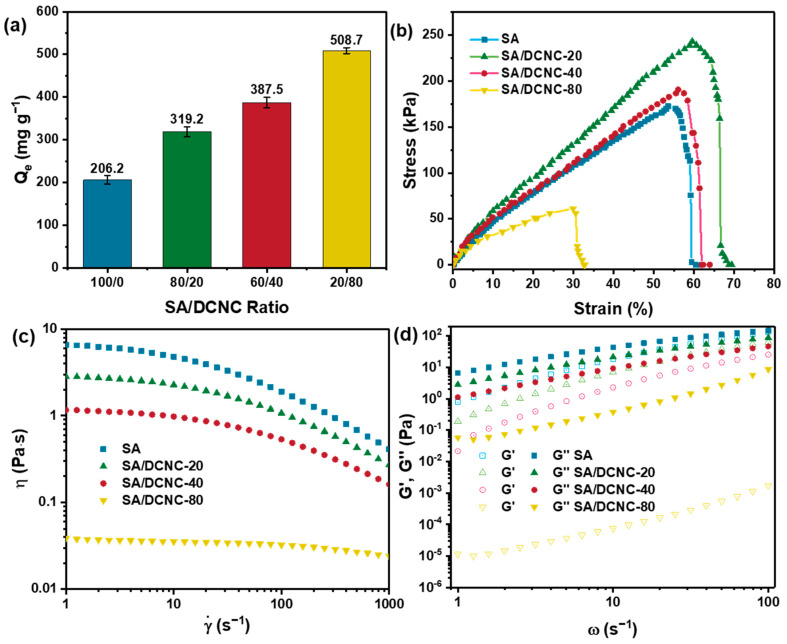
(**a**) Effect of SA–DNCN weight ratio on doxycycline adsorption efficiency (*C_0_* = 500 mg L^−1^, pH = 7, 24 h); (**b**) tensile stress–strain curves of hydrogel strip samples with different SA–DCNC composition ratios; (**c**) shear rate dependence of viscosity, and (**d**) angular frequency dependence of storage modulus (G′) and loss modulus (G″) for 2.5 wt.% of various SA–DCNC mixture suspensions.

**Figure 7 nanomaterials-13-01161-f007:**
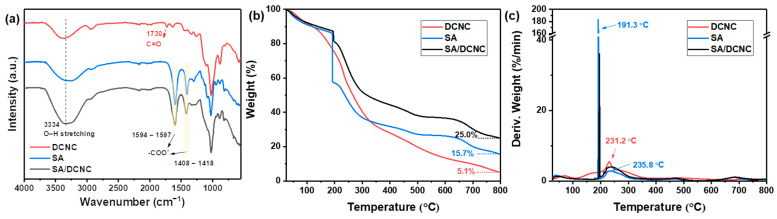
(**a**) FT-IR spectra, (**b**) TGA, and (**c**) DTG curves of SA, DCNC, and SA–DCNC-40.

**Figure 8 nanomaterials-13-01161-f008:**
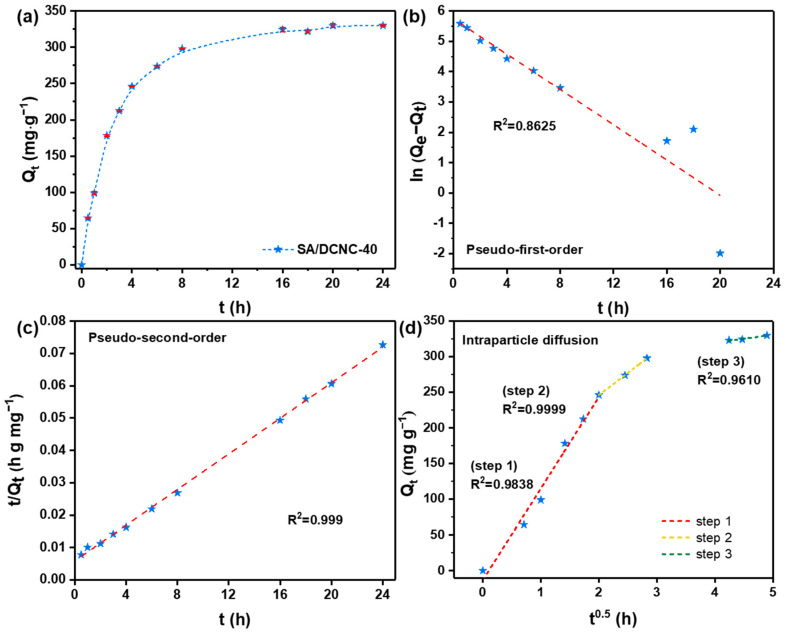
(**a**) Effect of contact time on the adsorption of doxycycline onto SA–DCNC-40 hydrogel beads and the kinetic model fittings: (**b**) pseudo-first-order, (**c**) pseudo-second-order, and (**d**) intraparticle diffusion.

**Figure 9 nanomaterials-13-01161-f009:**
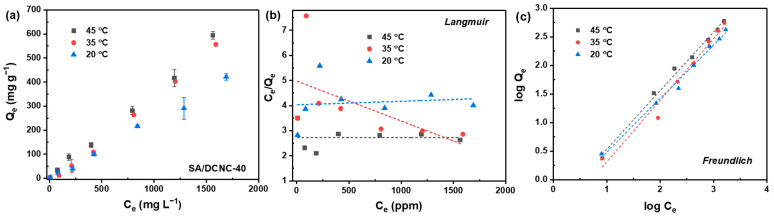
(**a**) Adsorption isotherm of DOXY on SA–DCNC-40 hydrogel beads at varying temperatures, and the data fitted by (**b**) Langmuir and (**c**) Freundlich isotherm models.

**Figure 10 nanomaterials-13-01161-f010:**
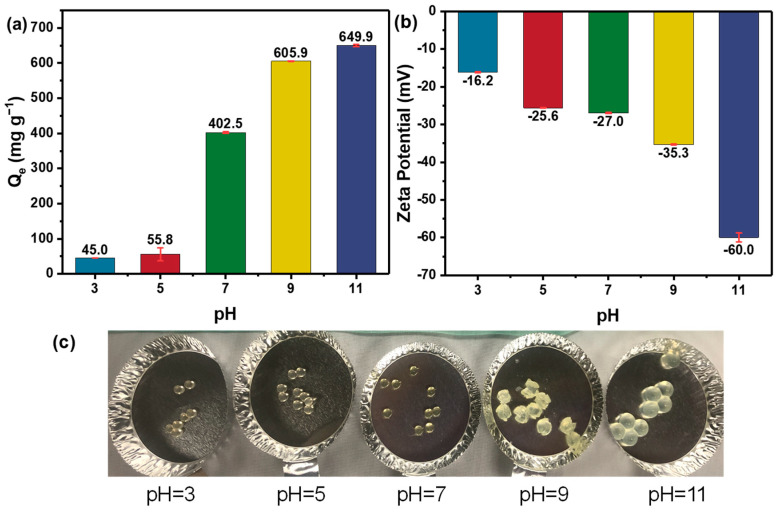
(**a**) Effect of pH on the adsorption for DOXY on the SA–DCNC-40 hydrogel beads, (**b**) zeta potential of SA–DCNC-40 mixed suspension, and (**c**) photographs of SA–DCNC-40 hydrogel beads after adsorption under different pH conditions.

**Figure 11 nanomaterials-13-01161-f011:**
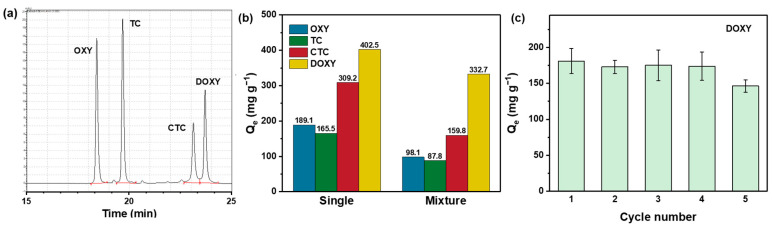
(**a**) HPLC chromatogram of 10 mg/L OXY, TC, CTC and DOXY standard solution; (**b**) OXY, TC, CTC, and DOXY single-drug adsorption and their competitive mixture adsorption by SA–DCNC-40 hydrogel beads (*C*_0_ = 500 mg L^−1^, pH = 7); (**c**) recyclability of SA–DCNC-40 hydrogel beads for DOXY adsorption.

**Figure 12 nanomaterials-13-01161-f012:**
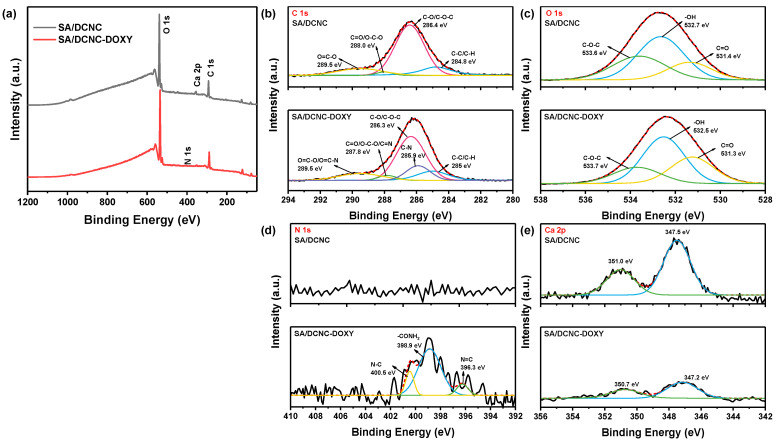
XPS spectra of SA–DCNC and SA–DCNC-DOXY: (**a**) wide scan and deconvolutions of (**b**) C 1s, (**c**) O 1s, (**d**) N 1s, and (**e**) Ca 2p.

**Figure 13 nanomaterials-13-01161-f013:**
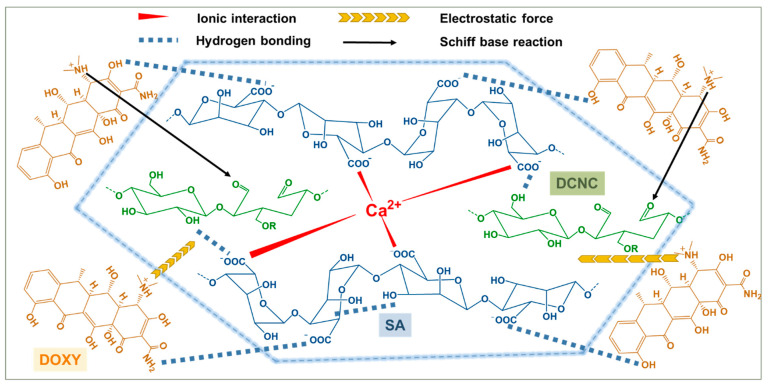
Schematic illustration of various proposed adsorption mechanisms between DOXY and SA–DCNC composite hydrogel beads.

**Table 1 nanomaterials-13-01161-t001:** Chemical structures and characteristics of the four selected tetracyclines.

Compound	Structural Formula	pKa_1_	pKa_2_	pKa_3_
Tetracycline (TC)	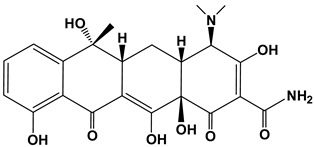	3.3	7.7	9.7
Oxytetracycline (OTC)	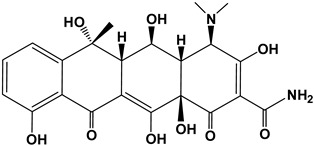	3.3	7.3	9.1
Chlortetracycline (CTC)	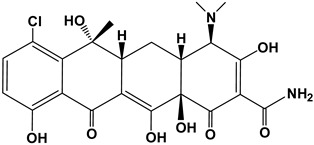	3.3	7.4	9.3
Doxycycline (DOXY)	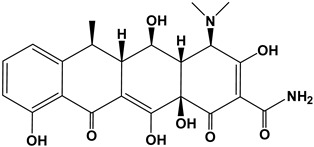	3.4	7.7	9.3

**Table 2 nanomaterials-13-01161-t002:** Composition, bead diameter (D_B_), BET specific surface area (SSA), pore volume (V_P_), and pore radius (r_P_) of various SA–DCNC samples.

Sample	SA–DCNC Weight Ratio	Component Weight			D_B_ (mm)	SSA (m^2^ g^−1^)	V_P_ (m^2^ g^−1^)	r_P_ (nm)
		SA (g)	DCNC (g)	Water (g)				
SA	100/0	0.75	0	29.25	2.8 ± 0.1	23.44	0.03	2.13
SA–DCNC-20	80/20	0.6	0.15	29.25	2.7 ± 0.1	35.43	0.05	1.90
SA–DCNC-40	60/40	0.45	0.3	29.25	2.6 ± 0.1	51.04	0.07	1.70
SA–DCNC-80	20/80	0.15	0.6	29.25	2.5 ± 0.2	46.47	0.04	1.70

**Table 3 nanomaterials-13-01161-t003:** Adsorption kinetic parameters for DOXY removal by SA–DCNC-40.

Kinetic Model	Parameters		
Experimental	*Q_e_*_,exp_ (mg g^−1^)		-
	329.9		-
Pseudo-first-order $\ln\left(Q_{e}-Q_{t}\right)=\ln Q_{e}-K_{1}t$	*Q_e_*_,1_ (mg g^−1^)	*K* _1_	R^2^
	310.7	0.29	0.863
Pseudo-second-order $\frac{t}{Q_{t}}=\frac{1}{Q_{e}}t+\frac{1}{K_{2}\;Q_{e}^{2}}$	*Q_e_*_,2_ (mg g^−1^)	*K* _2_	R^2^
	363.6	0.0013	0.999
Intraparticle diffusion $Q_{t}=K_{id}\;{\left(t\right)}^{0.5}+C_{i}$	*C_i_*	*K_id_*	R^2^
Step 1	−13.17	128.30	0.984
Step 2	121.21	62.43	1.000
Step 3	275.81	10.99	0.961

**Table 4 nanomaterials-13-01161-t004:** Freundlich adsorption isotherm parameters for DOXY removal by SA–DCNC-40 at different temperatures.

Isotherm Model	Parameters	Temperature		
		20 °C	35 °C	45 °C
Freundlich $\log Q_{e}=\log K_{F}+\frac{1}{n}\log C_{e}$	*K_F_*	0.24	0.36	0.59
	1/*n*	1.00	0.98	0.93
	R^2^	0.995	0.997	0.996

**Table 5 nanomaterials-13-01161-t005:** Comparison of the maximum adsorption capacities of different adsorbents for doxycycline removal (at 25 °C).

Adsorbents	*Q_m_* (mg g^−1^)	Condition	Refs
Cu(II)-impregnated biochar	52.4	25 °C, pH 8	[46]
Iron-loaded sludge biochar	129.0	20 °C, pH 6	[12]
Rice straw biochar	170.4	25 °C, pH 6	[47]
Kaolinite–CoFe_2_O_4_ nanoparticles	240	25 °C, pH 6	[44]
Graphene-like layered molybdenum disulfide	319	30 °C, pH 7	[48]
Graphene oxide	398.4	25 °C, pH 3.6	[49]
Carboxymethyl cellulose- and chitosan modified-magnetic alkaline Ca–bentonite	599	25 °C, pH 7	[9]
SA–DCNC composite hydrogel	421.5	25 °C, pH 7	This study

## Data Availability

Not applicable.

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
