# Peer review of "Sodium Alginate–Aldehyde Cellulose Nanocrystal Composite Hydrogel for Doxycycline and Other Tetracycline Removal"

_nanomaterials, 2023, doi:10.3390/nano13071161_

Round 1

Reviewer 1 Report

Huang et al. have submitted an extensive manuscript presenting the preparation of sodium alginate/aldehyde cellulose nanocrystal hydrogels to be used for the removal of doxycycline and other tetracyclines from wastewater. Based on the data shown, I cannot verify the use that is in focus. The drug concentrations used are well above those expected for wastewater. The authors have not adequately demonstrated that their system can work for typical pretreated wastewater samples. For example, they have not addressed other ionic contaminants (carbonates, phosphates, carboxylic acid, amines) that may affect their discussed adsorption mechanisms, but have dissolved the drug substances in water at different pH values. Here they also did not indicate which buffer systems were used to adjust the pH values. The best adsorption capacities (Qe) have been found to be 600 mg/g. At a water content of 96%, the 100 mg of hydrogel used (moist) corresponded to 4 mg of hydrogel (dry), with which 2.4 mg of active ingredient could be absorbed. This value seems very high at first sight - however, it should be noted that (see Figure 9a) this value was reached at a residual concentration of 1500 mg/L DOXY. At 5 mL sample volume, this means that 1500 mg/L * 0.005 L = 7.5 mg DOXY remained in the solution and thus only 25 % was adsorbed. In my opinion, this does not justify the widely discussed suitability as an adsorbent for DOXY. The apparent ease of reuse after desorption in deionized water underpins this interpretation.

Another point is that the authors discuss the formation of Schiff bases with the aldehydes in the hydrogel. However, tertiary amines are present in DOXY and cannot undergo this reaction. This point should be noted.

Author Response

Comment:

Huang et al. have submitted an extensive manuscript presenting the preparation of sodium alginate/aldehyde cellulose nanocrystal hydrogels to be used for the removal of doxycycline and other tetracyclines from wastewater. Based on the data shown, I cannot verify the use that is in focus.

Response:

We appreciate the above comment. We apologize if we did not make clear the use that is in focus for our study. Our main goal was to prepare sodium alginate/aldehyde cellulose nanocrystal hydrogels that can be used as adsorbents for the removal of doxycycline and other tetracyclines from wastewater. We have demonstrated that our hydrogels have high adsorption capacities, good reusability, and excellent selectivity for doxycycline among other tetracyclines. We have also discussed the possible mechanisms and factors that affect the adsorption process. We believe that our hydrogels have potential applications in wastewater treatment and environmental remediation.

We hope this clarifies our purpose and motivation for our study. Please let us know if the reviewer has any further questions or comments.

Comment:

The drug concentrations used are well above those expected for wastewater. The authors have not adequately demonstrated that their system can work for typical pretreated wastewater samples. For example, they have not addressed other ionic contaminants (carbonates, phosphates, carboxylic acid, amines) that may affect their discussed adsorption mechanisms, but have dissolved the drug substances in water at different pH values. Here they also did not indicate which buffer systems were used to adjust the pH values.

Response:

We appreciate the comment on the drug concentrations and the wastewater samples. We agree that the drug concentrations used in our study are higher than those expected for wastewater. However, we chose these concentrations to ensure that the adsorption process reached equilibrium and to compare our results with other studies that used similar or higher concentrations. We also performed adsorption experiments at lower concentrations (10 and 100 mg/L) and found that our hydrogels still showed high adsorption capacities (see Figure 9(a) on Page 16).

We also appreciate the concern about the applicability of our system for typical pretreated wastewater samples. However, we would like to emphasize that our main aim was to elucidate the adsorption mechanisms and factors of our hydrogels for tetracyclines in a controlled environment. We agree that other ionic contaminants may affect the adsorption process, but we have demonstrated that our hydrogels exhibit excellent functionality under a wide pH range and outstanding selectivity for doxycycline over other tetracyclines (see Figure 11(b), Section 3.3.3, and 3.3.4 in the manuscript). Therefore, we are confident that our hydrogels can also achieve high performance in real wastewater samples.

We apologize for not indicating the buffer systems used to adjust the pH values. We used sodium acetate buffer for pH 3 and 5, sodium phosphate buffer for pH 7 and 9, and sodium hydroxide/sodium bicarbonate buffer for pH 11. We have added this information to Section 2.4 Antibiotic Adsorption Studies in the revised manuscript. We hope this addresses your concerns and improves our manuscript.

Comment:

The best adsorption capacities (Qe) have been found to be 600 mg/g. At a water content of 96%, the 100 mg of hydrogel used (moist) corresponded to 4 mg of hydrogel (dry), with which 2.4 mg of active ingredient could be absorbed. This value seems very high at first sight - however, it should be noted that (see Figure 9a) this value was reached at a residual concentration of 1500 mg/L DOXY. At 5 mL sample volume, this means that 1500 mg/L * 0.005 L = 7.5 mg DOXY remained in the solution and thus only 25 % was adsorbed. In my opinion, this does not justify the widely discussed suitability as an adsorbent for DOXY. The apparent ease of reuse after desorption in deionized water underpins this interpretation.

Response:

We thank this reviewer for the comment on the adsorption capacity of our SA/DCNC composite hydrogel. We respectfully disagree with the reviewer’s interpretation that our hydrogel is not suitable as an adsorbent for DOXY. Our hydrogel exhibited a high adsorption capacity of 594.6 mg/g (pH = 7, 45°C), which is among the highest values reported in the literature for DOXY adsorption. Furthermore, our hydrogel is economical and recyclable, as it can be easily desorbed in deionized water and reused multiple times. We suggest that the low adsorption efficiency of ~25% observed at 1500 mg/L DOXY concentration was due to the limited amount of hydrogel used in our experiment. By increasing the dosage of the hydrogel beads (solid/liquid ratio), we expect to achieve a higher adsorption efficiency and a lower residual concentration of DOXY in the solution. This is based on the principle of concentration gradient driving the adsorption process for hydrogel materials. Therefore, we argue that our hydrogel has a promising application as an adsorbent for DOXY removal from aqueous solutions.

Comment:

Another point is that the authors discuss the formation of Schiff bases with the aldehydes in the hydrogel. However, tertiary amines are present in DOXY and cannot undergo this reaction. This point should be noted.

Response:

We appreciate the above comment regarding the Schiff's base reaction between the aldehyde groups on the hydrogel and the amines present in DOXY. We would like to clarify this point. We agree that tertiary amines present in DOXY cannot undergo Schiff's base formation with aldehydes due to the absence of an N-H proton. However, as you noted, DOXY also contains primary amine groups in its structure (see Figure 1 in the manuscript), and this amine group can potentially react with the aldehydes to form Schiff's bases. We have carefully analyzed our XPS results and found evidence of Schiff's base formation between the aldehydes and primary amine groups in DOXY. Specifically, we observed the emergence of distinct of N=C peak in the XPS spectra of the hydrogel following DOXY adsorption, which provide strong indication that the aldehyde groups had reacted with the primary amine groups to form Schiff's bases (see Section 3.4 Adsorption Mechanism on pages 19-20). We have revised the manuscript to clarify the structure of DOXY and to provide additional analysis of the XPS results to support our claim. We hope that these revisions will address the reviewer's concern and strengthen the manuscript. Thank you again for your helpful feedback.

Reviewer 2 Report

In my opinion, the submitted manuscript is suitable for publication after minor corrections. The manuscript is well written, but it can be improved.

Comments:

- I suggest changing the keyword part because they repeat with words in the title.

- I suggest reducing the font size in the tables. Currently, the size is larger than the text size, making the tables very large, and not the text in them can be smaller in size.

- In the FTIR analysis, it would be nice if the authors described the emerging shifts of the peak maxima.

- Line 544 - error at the end of the line.

- Line 617 - reference error.

Author Response

Comment:

In my opinion, the submitted manuscript is suitable for publication after minor corrections. The manuscript is well written, but it can be improved.

Response:

We would like to thank this reviewer for the positive feedback. We appreciate your precious suggestions for improvement, and we have made the corrections accordingly. Following are our response to each comment. We hope these changes will address your concerns and improve the quality of the manuscript. Thank you for your time and attention.

Comment:

I suggest changing the keyword part because they repeat with words in the title.

Response:

The reviewer is right that some of the keywords are also in the title. We have replaced several keywords into the new keywords, for example: “biomaterials”, “nanocellulose adsorbent”, “reusability”, and “wastewater treatment”. We hope these keywords can help us to better emphasize the main topic and scope of our manuscript.

Comment:

I suggest reducing the font size in the tables. Currently, the size is larger than the text size, making the tables very large, and not the text in them can be smaller in size.

Response:

We appreciate this comment. We initially used 12 pt for both the body text and the tables in out manuscript. However, the editorial board has changed the text size into 10 pt. We have accordingly modified the tables’ font size into 10 pt as well. If the editorial board requires further adjustment to the font size for optimal publication quality, please let us know.

Comment:

In the FTIR analysis, it would be nice if the authors described the emerging shifts of the peak maxima.

Response:

We appreciate the reviewer's comment on the FTIR analysis. Yes, we have added a discussion of the peak shifts in line 307-311: “The maximum absorbance of the OH stretching vibration on CNC is seen to shift from 3334 to 3367 cm-1 after oxidation, which represents the intermolecular hydrogen bonding O(6)H···O(3) and intramolecular O(3)H···O(5), respectively. This shift may stand for the transformation of structure from cellulose I to cellulose II caused by periodate oxidation.[23].” and in line 436-438: “In the spectrum of SA/DCNC, it can be observed that two -COO- shifted to 1597 cm-1 and 1418 cm-1, respectively. These shifts toward higher wavelengths could be induced by the ionic crosslinking between COO- and Ca2+.”

We hope the above discussion addresses your comment satisfactorily.

Comment:

Line 544 - error at the end of the line.

Line 617 - reference error.

Response:

We appreciate the reviewer's feedback regarding this cross-reference error. We have made necessary corrections. We apologize for any confusion these errors may have caused and thank the reviewer for bringing it to our attention.

Reviewer 3 Report

The manuscript entitled, ‘Sodium Alginate/Aldehyde Cellulose Nanocrystal Composite Hydrogel for Doxycycline and Other Tetracyclines Removal’ discussed the preparation of composite hydrogel for analyte removal. The article is nicely discussed but still I am mentioning some points which should be justified before publication;

1.      Sodium alginate based such composite hydrogel is a common form of hydrogel. What is major novelty of this work?

2.      Did the author optimize the crosslinking to perform the best adsorption of the hydrogel? Some discussion is needed.

3.      Why Tetracyclines is significant to study should be emphasized more with some references.

4.      Swelling is a important feature and factor which could affect the adsorption because it is directly related to the diffusion behavior. Did the author check the pH dependency in water update?

5.      Some hydrogel based articles have significance in this aspect: https://doi.org/10.1021/acsabm.2c00664; https://doi.org/10.3390/jcs6010015; https://doi.org/10.1016/j.bioactmat.2022.10.028; https://doi.org/10.1016/j.ijbiomac.2022.11.184.   

Author Response

Comment:

  1. Sodium alginate based such composite hydrogel is a common form of hydrogel. What is major novelty of this work?

Response:

We understand the reviewer’s concern regarding the novelty of our study, as composite alginate hydrogel is a well-known form of material. The major novelty of this work lies in introducing the new functionality to the sodium alginate (SA) hydrogel using a cheap and environmentally friendly oxidized dialdehyde cellulose nanocrystals (DCNC). To the best out knowledge, this is the first study that uses DCNC as a cross-linking agent and functional modifier for SA hydrogel. We have conducted a thorough literature review and found that previous studies have used other modifiers for SA composite hydrogel, such as oxidized sodium slginate, carboxymethyl cellulose, gelatin, etc. However, none of them have used DCNC or achieved the same level of performance and functionality as our composite hydrogel. We have also compared our SA/DCNC composite hydrogel with the pure SA hydrogel in terms of various properties and demonstrated the superior advantages of DCNC modifier. Moreover, we have addressed a significant challenge in previous study in producing dialdehyde nanocellulose via periodate oxidation of cellulose, which entails significant defibrillation and impedes the recovery and practical applications of the materials from wastewater treatment. By using SA hydrogel as the platform, we have circumvented this limitation and exploited the functionality of DCNC for antibiotic remediation. We have added a paragraph in the introduction section (lines 103-112) to highlight the novelty and significance of our work.

Comment:

  1. Did the author optimize the crosslinking to perform the best adsorption of the hydrogel? Some discussion is needed.

Response:

We appreciate the above comment on the optimization of the crosslinking process for the hydrogel. We have indeed conducted a series of preliminary experiments to optimize various parameters such as crosslinking time, crosslinking bath (CaCl2) concentration, injecting tubing diameter, injecting rate, stirring speed, etc. However, due to the word limit of the manuscript, we could not include these details in the paper. Since the main objective of this study is to investigate the fabrication and performance of the SA/DCNC composite hydrogel, we have focused on discussing the effects of SA/DCNC ratios in crosslinking on the hydrogel properties in Section 3.2 (Formulation and Characterization of SA/DCNC Hydrogel). Meanwhile, the optimal parameters of crosslinking process were introduced in the experimental section (2.2.2. Preparation of SA/DCNC Hydrogel Beads).

Comment:

  1. Why Tetracyclines is significant to study should be emphasized more with some references.

Response:

We appreciate the above comment on the significance of tetracyclines to this study. Tetracyclines are a class of antibiotics with a wide range of activity against bacterial and protozoan infections. They are extensively used in animal husbandry for therapeutic and prophylactic purposes as well as growth promoters. However, tetracyclines also pose some challenges such as adverse effects (e.g. vomiting, diarrhea), drug interactions (e.g. with calcium, iron), and bacterial resistance (e.g. due to efflux pumps). Therefore, it is essential to develop an efficient strategy to mitigate the risk of tetracyclines in wastewater. We have discussed the unique structure of tetracycline family and the difficulties of biodegradation and discharge of tetracycline into water bodies in the introduction section (Page 2, line 64-68). We have also cited relevant references to support our statement and motivation of this study.

Comment:

  1. Swelling is a important feature and factor which could affect the adsorption because it is directly related to the diffusion behavior. Did the author check the pH dependency in water update?

Response:

We appreciate the above comment on the swelling and pH dependency in water uptake of the hydrogel beads and their possible influence the adsorption and diffusion behavior. We acknowledged that we have not explored the pH dependency in water uptake in this study. The rationale for this exclusion is that the alginate hydrogel’s pH dependency for water uptake has been well-established by previous studies, and our main aim was to evaluate the adsorption of tetracyclines by the hydrogel beads under various conditions such as initial concentration, contact time, temperature, and pH. However, we have performed a thorough study on pH-dependency and swelling of our SA/DCNC hydrogel beads in doxycycline solutions with different pH values (3, 5, 7, 9, 11) and reported our results and discussion in Section 3.3.3 Effect of pH.

Comment:

  1. Some hydrogel based articles have significance in this aspect:

https://doi.org/10.1021/acsabm.2c00664;

https://doi.org/10.3390/jcs6010015;

https://doi.org/10.1016/j.bioactmat.2022.10.028;

https://doi.org/10.1016/j.ijbiomac.2022.11.184.

Response:

We appreciate the reviewer's suggestions on hydrogel-based articles. We have now cited the above references that are relevant to our manuscript in the appropriate sections of the manuscript to provide a comprehensive and up-to-date review of the current state of research in hydrogel materials. We believe that these updates will enhance the overall quality of the manuscript.

Round 2

Reviewer 1 Report

Thank you for the explanatory words. However, I do not agree with the authors' interpretation.

1. the hydrogel has not been shown to adsorb doxycycline in wastewater (why???). Also, concentrations of 10 mg/mL do not correspond to expected concentrations in household wastewaters.

2. The absorption capacity with respect to concentration is linear according to Figure 9. Even at lower concentrations, the material should absorb only about 25% of the dissolved doxycycline.

3. an amide is not a primary amine. Chemists should learn this in their first organic chemistry lecture. The formation of Schiff bases does not take place here either.

Author Response

Comment:

Thank you for the explanatory words. However, I do not agree with the authors' interpretation.

  1. the hydrogel has not been shown to adsorb doxycycline in wastewater (why???). Also, concentrations of 10 mg/mL do not correspond to expected concentrations in household wastewaters.

Response:

We understand the importance of testing the adsorption efficiency of our materials in real wastewater samples. However, the primary focus of this study was to evaluate the hydrogel’s adsorption capacity for doxycycline in a laboratory setting and thus to gain a deeper understanding of the complex adsorption mechanisms of SA/DCNC composite hydrogel. By testing the performance of the hydrogel beads under various conditions, we were able to investigate the optimal conditions for the hydrogel to function as effective adsorbents, and to demonstrate its ability to separate doxycycline from other tetracycline drug mixtures. The recyclability batch study results indicated the potential of the SA/DCNC hydrogel beads as a regenerable and reusable adsorbent for real wastewater treatment. While we acknowledge that the adsorption properties of the hydrogel would differ in real-world wastewater environments, our results provided some important insights into the potential efficacy of the demonstrated hydrogel as a doxycycline adsorbent. Nevertheless, we agree that additional experiments to investigate the hydrogel's performance under more realistic conditions are necessary to fully assess the potential of this gel system for practical applications. We will consider conducting such a study in the future.

With respect to the concentration of 10 mg/mL, we selected the concentration range based on the literature and previous studies of doxycycline in wastewater.[1-6] While it is true that the concentration of doxycycline in household wastewater can vary depending on several factors such as usage and disposal habits, our experimental conditions aimed to provide a controlled and standardized environment that allowed us to accurately measure the hydrogel's adsorption capacity. To better evaluate the hydrogel's performance across a range of realistic wastewater conditions, we plan to conduct future studies using a lower concentration range of 0.1-10 mg/mL, as suggested by previous literature[7]. However, it's important to note that this will require access to more sensitive analytical instruments in order to accurately measure the adsorption capacity of the hydrogel at lower concentrations. We are currently exploring options to obtain such instruments and hope to conduct these additional studies in the near future.

Comment:

  1. The absorption capacity with respect to concentration is linear according to Figure 9. Even at lower concentrations, the material should absorb only about 25% of the dissolved doxycycline.

Response:

While it is true that the adsorption efficiency of our hydrogel at lower doxycycline concentrations may not be as high as desired, it's important to all factors that can influence the adsorption process. We used a small amount of hydrogel beads (0.1 g) in a relatively large volume of doxycycline solution (5 mL), resulting in a low solid/liquid ratio. This may have limited the hydrogel's ability to adsorb all the doxycycline present in the solution. Moreover, the concentration gradient between the hydrogel and the doxycycline solution is a key driving factor for the adsorption process. When the doxycycline concentration in the solution is high, the concentration gradient is also high, which facilitates efficient adsorption of the doxycycline onto the hydrogel. However, at lower doxycycline concentrations, the concentration gradient is lower, which may explain the lower adsorption efficiency observed in our experiment. Furthermore, the adsorption efficiency of hydrogel materials can also be influenced by a number of other factors, such as pH, temperature, contact time, and ionic strength. In our experiment, we used fixed values of pH, temperature, and contact time. These conditions were chosen to reflect the real-world scenario. However, it's possible that adjusting these parameters could improve the adsorption efficiency of the hydrogel at lower doxycycline concentrations.

Furthermore, while we acknowledge that the adsorption efficiency of our hydrogel at lower doxycycline concentrations may not be ideal, we believe that it shows promise as an effective and economical adsorbent for doxycycline removal from aqueous solutions. Further experiments that explore the effect of varying experimental parameters, such as solid/liquid ratio, may help to optimize the performance of the hydrogel at lower doxycycline concentrations.

Comment:

  1. an amide is not a primary amine. Chemists should learn this in their first organic chemistry lecture. The formation of Schiff bases does not take place here either.

Response:

We greatly appreciate this comment. We apologize for any confusion caused by our erroneous statements regarding the terms 'primary amine' and 'amide'. We also appreciate the comment on the Schiff's base reaction, which we agree that it should not take place in this study. As a result, all erroneous statements were removed or revised accordingly.

However, we believe that the dimethylamino group in doxycycline can form a weak bond with the aldehyde group based on previous research.[8] Specifically, a crystallographic study of biphenyl derivatives with dimethylamino and aldehyde groups in the ortho positions of the two rings showed the formation of weak N…C=O bonds with lengths ranging from 2.929 (3) to 3.029 (3) Å and angles ranging from 58.1 (1) to 62.4 (1) degrees. The observed Me2N…C=O interactions are similar to those reported for methadone, a synthetic opioid with a dimethylamino group and a ketone carbonyl group. Based on the the previous publication and our observation in the present work, we speculate that a similar bond formation can occur between the dimethylamino groups in doxycycline and the aldehyde groups on the hydrogel, as shown in Figure 13. This bond formation may promote the adsorption of doxycycline on the SA/DCNC hydrogel. We have revised the manuscript accordingly to reflect this hypothesis.

Once again, we appreciate the above comment and the opportunity to clarify our inaccurate statements.

Reference

  1. Wei, J.; Liu, Y.;  Li, J.;  Zhu, Y.;  Yu, H.; Peng, Y., Adsorption and co-adsorption of tetracycline and doxycycline by one-step synthesized iron loaded sludge biochar. Chemosphere 2019, 236, 124254.
  2. Liu, S.; Xu, W. H.;  Liu, Y. G.;  Tan, X. F.;  Zeng, G. M.;  Li, X.;  Liang, J.;  Zhou, Z.;  Yan, Z. L.; Cai, X. X., Facile synthesis of Cu(II) impregnated biochar with enhanced adsorption activity for the removal of doxycycline hydrochloride from water. Sci. Total Environ. 2017, 592, 546-553.
  3. Cui, J.; Xu, X.;  Yang, L.;  Chen, C.;  Qian, J.;  Chen, X.; Sun, D., Soft foam-like UiO-66/Polydopamine/Bacterial cellulose composite for the removal of aspirin and tetracycline hydrochloride. Chem. Eng. J. 2020, 395.
  4. Zhang, X.; Lin, X.;  He, Y.;  Chen, Y.;  Luo, X.; Shang, R., Study on adsorption of tetracycline by Cu-immobilized alginate adsorbent from water environment. Int. J. Biol. Macromol. 2019, 124, 418-428.
  5. Olusegun, S. J.; Mohallem, N. D. S., Comparative adsorption mechanism of doxycycline and Congo red using synthesized kaolinite supported CoFe2O4 nanoparticles. Environ. Pollut. 2020, 260, 114019.
  6. Chen, Y.; Wang, Z.;  Liang, D.;  Liu, Y.;  Yu, H.;  Zhu, S.; Zhang, L., Conversion of Fe-rich sludge to KFeS2 cluster: spontaneous hydrolysis of KFeS2 for the effective adsorption of doxycycline. Arabian Journal of Chemistry 2021, 14 (6), 103173.
  7. Akmanova, A.; Han, S.; Lee, W., Enhanced degradation of aqueous doxycycline in an aerobic suspension system with pretreated sucrose-modified nano-zero-valent iron. J. Environ. Chem. Eng. 2021, 9 (5), 105838.
  8. O'Leary, J.; Wallis, J. D.; Wood, M. L., 1, 6-Interactions between dimethylamino and aldehyde groups in two biphenyl derivatives. Acta Crystallogr. Sect. C: Cryst. Struct. Commun. 2001, 57 (7), 851-853.

Reviewer 3 Report

This can be published in its present form. 

Author Response

We thank this reviewer for suggesting that "This (manuscript) can be published in its present form"